# Precision and reliability of tape measurements in the assessment of head and neck lymphedema

**Adit Chotipanich** [1] *, **Nampheng Kongpit** [2]

**1** Department of Medical Services, Head and Neck Unit, Chonburi Cancer Hospital, Ministry of Public Health, Chonburi, Thailand, **2** Department of Medical Services, Nursing Unit, Chonburi Cancer Hospital, Ministry of Public Health, Chonburi, Thailand

* adit_c@ccc.in.th

## Abstract

**Data Availability Statement:** All relevant data are within the manuscript and its Supporting Information files.

**Funding:** The author(s) received no specific funding for this work.

### Objectives

Tape measurement is a commonly used method in the clinical assessment of lymphedema. However, few studies have assessed the precision and reliability of tape measurement in assessing head and neck lymphedema. This study aimed to evaluate the reliability and precision of using tape measurement, performed by different evaluators, for the assessment of head and neck lymphedema.

### Methods

This study was conducted at a tertiary care cancer hospital. Between January and December 2019, 50 patients with head and neck cancers and 50 normal subjects were enrolled. Each subject was examined using tape measurements for 7 point-to-point distances of facial landmarks, 3 circumferences of the neck (upper, middle, and lower), and 2 circumferences of the face (vertical and oblique) by 3 random examiners. Test precision and reliability were assessed with the within-subject standard deviation ($S_w$) and intra-class correlation coefficient (ICC), respectively.

### Results

Overall, the standard deviation of the tape measurements varied in the range of 4.6 mm to 18.3 mm. The measurement of distance between the tragus and mouth angle ($S_w$: 4.6 mm) yielded the highest precision, but the reliability (ICC: 0.66) was moderate. The reliabilities of neck circumference measurements (ICC: 0.90–0.95) were good to excellent, but the precisions ($S_w$: 8.3–12.3 mm) were lower than those of point-to-point facial measurements ($S_w$: 4.6–8.8 mm).

**Competing interests:** The authors have declared that no competing interests exist.

## Conclusions

The different methods of tape measurements varied in precision and reliability. Thus, clinicians should not rely on a single measurement when evaluating head and neck lymphedema.

## Introduction

Head and neck lymphedema is a condition commonly found in patients after head and neck cancer treatments [1]. The accurate assessment of head and neck lymphedema can help clinicians determine the best treatment option in these patients. Several assessment methods have been proposed, which can be categorized into two groups. The first group comprises of qualitative rating scales using clinical signs and symptoms. The second group involves several quantitative methods, such as measuring the distance of the head and neck anatomy with a tape, tissue-thickness measurements using computed tomography (CT) or ultrasound images, and bioelectric impedance analysis [2].

Diagnosis and evaluation of head and neck lymphedema are based on measuring the change from baseline. These usually involve taking the measurements at different times, while being performed by different evaluators. A minor discrepancy between the measurements could affect the reliability of the clinician's judgment. Tape measurement remains a widely used method in the clinical assessment of lymphedema. However, there are few studies on the precision and reliability of tape measurement in head and neck lymphedema. The objective of this study was to evaluate the precision and reliability of tape measurement techniques performed by different evaluators in the assessment of head and neck lymphedema.

## Materials and methods

### Study design

This cross-sectional study was performed at the Chonburi Cancer Hospital. Healthy subjects and patients with head and neck cancers who previously received treatment were non-randomly enrolled between January 2019 and December 2019. The inclusion criteria were as follows: age ≥18 years and those who are able to sit upright to complete the examination. Subjects who had severe deformities of the head and neck were excluded. The study's protocol was approved by the Chonburi Cancer Hospital ethics committee, and all participants provided written informed consent prior to their enrollment.

### Evaluator recruitment and measurement procedures

The evaluators included 20 physicians and nurses who were trained to use tape measurements. The evaluators had at least 2 years of experience working with patients in the head and neck clinic. All evaluators successfully completed practice measurement training with the principle researchers before performing the measurements on the study subjects. Of these 20 evaluators, 3 were randomly assigned to perform the measurements on each enrolled subject in a consistent setting, and the measurements were conducted consecutively. The same soft vinyl medical measuring tape (MABIS®) was used to perform all measurements. Each evaluator was blinded to the results of the other evaluators. The following measurements were taken by the evaluators: 7 key facial distances, 2 facial circumferences, and 3 neck circumferences (upper, middle, and lower) (Fig 1). Measured distances between key facial landmarks were taken on

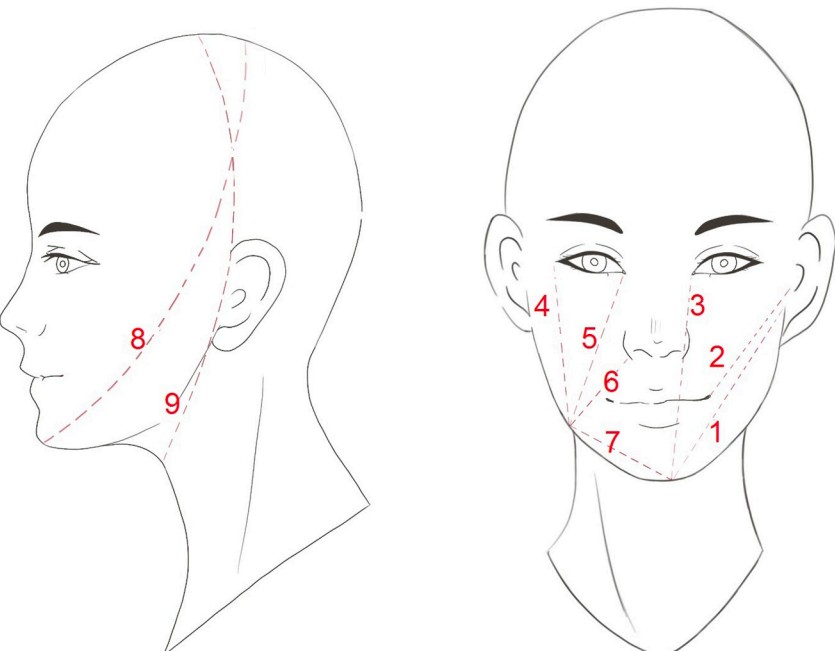

**Fig 1. The tape placement lines.** (1) Tragus to mental protuberance, (2) tragus to mouth angle, (3) mental protuberance to internal eye corner, (4) mandibular angle to external eye corner, (5) mandibular angle to internal eye corner, (6) mandibular angle to nasal wing, (7) mandibular angle to mental protuberance, (8) diagonal facial circumference—chin to crown of the head, and (9) vertical facial circumference—in front of the ear. The lines used for measuring the neck circumferences are not shown in this figure.

the right side of the face of each subject. The landmarks of the upper and lower neck circumferences were the levels just above the hyoid bone and the clavicular head, respectively. The mid-neck circumference was estimated at the level halfway between the upper and lower neck circumferences.

## Statistical analysis

Data were analyzed using SPSS software (version 17.0; SPSS Inc., Chicago, IL, USA). Test precision was assessed with the within-subject standard deviation. A one-way analysis of variance (ANOVA) was used for calculating the within-subject standard deviation ($S_w$). The intraclass correlation coefficient (ICC) was used as a reliability index in interrater reliability analyses. We employed a single-measurement, absolute agreement, and two-way random effects model in the ICC analysis. The 95% confidence interval for the ICC was used as the basis for evaluating the level of reliability.

## Results

Between January and December 2019, 50 patients with head and neck cancers and 50 normal subjects were enrolled. The characteristics of the cancer patients are shown in Table 1. The analysis of the precision and reliability of the 12 types of tape measurements are shown in Table 2.

We used the 95% confidence interval of the ICC (95%CI) as the basis for evaluating the level of reliability using the following general guideline. Values less than 0.5 are indicative of poor reliability, values between 0.5 and 0.75 indicate moderate reliability, values between 0.75 and 0.9 indicate good reliability, and values greater than 0.90 indicate excellent reliability [3].

**Table 1. Characteristics of patients with cancer.**

| Patient and tumor characteristics | Number (%) |
|---|---|
| Sex | 10 women, 40 men |
| Diagnosis: | |
| Nasopharyngeal cancer | 3 (6%) |
| Oral and oropharyngeal cancers | 19 (38%) |
| Laryngeal and hypopharyngeal cancers | 23 (46%) |
| Others | 5 (10%) |
| Stage: III | 15 (30%) |
| IV | 35 (70%) |
| Lymph node status: | |
| N0 | 17 (34%) |
| N1 | 10 (20%) |
| N2 | 21 (42%) |
| N3 | 2 (4%) |
| Treatments | |
| Surgery | 44 (88%) |
| Without neck dissection | 9 (18%) |
| With bilateral neck dissection | 10 (20%) |
| With unilateral neck dissection | 25 (50%) |
| Radiation | 49 (98%) |
| Post-operative radiation (with or without chemotherapy) | 43 (86%) |
| Concurrent chemo-radiation | 6 (12%) |

Values are presented as numbers and percentages.

**Table 2. Analysis of the precision and reliability of the tape measurements.**

| Measurements | Range (mm.) | MSB (mm.) | MSW (mm.) | $S_w$ (mm.) | ICC | 95%CI | Reliability |
|---|---|---|---|---|---|---|---|
| Key facial distance measurements | | | | | | | |
| Tragus to mental protuberance | 130–183 | 226.9 | 28.2 | 5.3 | 0.70 | 0.61–0.78 | Moderate to good |
| Tragus to mouth angle | 95–140 | 142.6 | 20.8 | 4.6 | 0.66 | 0.56–0.74 | Moderate |
| Mandibular angle to nasal wing | 75–152 | 253.3 | 41.1 | 6.4 | 0.63 | 0.53–0.72 | Moderate |
| Mandibular angle to internal eye corner | 105–165 | 222.1 | 43.8 | 6.6 | 0.58 | 0.47–0.68 | Poor to moderate |
| Mandibular angle to external eye corner | 85–138 | 159.1 | 38.2 | 6.2 | 0.52 | 0.40–0.62 | Poor to moderate |
| Mental protuberance to internal eye corner | 92–144 | 155.8 | 29.1 | 5.4 | 0.59 | 0.49–0.69 | Poor to moderate |
| Mandibular angle to mental protuberance | 75–150 | 193.6 | 78.1 | 8.8 | 0.33 | 0.20–0.46 | Poor |
| Facial circumferences | | | | | | | |
| Diagonal: chin to crown of the head | 595–745 | 1953.7 | 138.6 | 11.8 | 0.81 | 0.75–0.86 | Good |
| Vertical: in front of the ears | 550–890 | 2655.0 | 335.9 | 18.3 | 0.70 | 0.61–0.77 | Moderate to good |
| Neck circumferences | | | | | | | |
| Superior neck | 250–505 | 5479.7 | 112.4 | 10.6 | 0.94 | 0.92–0.96 | Excellence |
| Middle neck | 242–440 | 3855.0 | 69.3 | 8.3 | 0.95 | 0.93–0.96 | Excellence |
| Inferior neck | 255–510 | 4075.5 | 152.2 | 12.3 | 0.90 | 0.86–0.93 | Good to excellence |
| Combination of measurements | | | | | | | |
| Tragus to mouth angle + middle neck | 337–570 | 5040.5 | 92.6 | 9.6 | 0.95 | 0.93–0.96 | Excellence |

MSB, mean square between groups; MSW, mean square within groups; $S_w$, within-subject standard deviation (reproducibility); ICC, intraclass correlation coefficient; 95%C, 95% confidence interval for the intraclass correlation

Overall, the standard deviation of the tape measurements varied in the range of 4.6 mm to 18.3 mm. Measuring the distance between the tragus and mouth angle yielded the highest precision ($S_w$: 4.6 mm), but the reliability was moderate (95%CI: 0.56–0.74). Measuring the mid-neck circumference had excellent reliability (95%CI: 0.93–0.96), but its precision was relatively moderate ($S_w$: 8.3 mm). Measuring the vertical facial circumference yielded the lowest precision ($S_w$: 18.3 mm), while measuring the distance from the mandibular angle to the mental protuberance yielded the lowest reliability (95%CI: 0.20–0.46).

## Discussion

Currently, there is no gold standard method for the measurement of head and neck lymphedema [2]. Although several novel measurement methods such as bio-impedance analysis [4], digital photograph analysis [5], and CT/MRI/Ultrasound imaging measurements [6] have been proposed, these methods are still not widely used in clinical settings. Therefore, the traditional technique, using a tape measure, remains an important tool for assessing lymphedema in various organs. However, studies assessing the precision and reliability of using a tape measurement for the head and neck are lacking. Thus, this study aimed to answer this question.

The M.D. Anderson Cancer Center head and neck lymphedema program has proposed a protocol for evaluating head and neck lymphedema, which includes patient interviews, photography, tape measurement, and staging of the edema to characterize the overall appearance and severity of the lymphedema [7]. Of these, tape measurement is the sole quantitative measurement. The tape measurement of head and neck lymphedema involves measuring the distance between prominent facial landmarks and circumferences of the face and neck. Diagnosis and progress evaluation of lymphedema rely on changes in these measures from baseline as well as the appearance of patients.

This study showed the varying precision and reliability of performing various tape measurements. The measurement of various distances between key facial landmarks yielded a relatively good precision, but the reliability was relatively low, while almost all circumferential measurements yielded good reliability and poor precision. The study also found that precision was generally greater when the measurement distance was shorter.

Similar to other studies [8, 9], we found a high degree of reliability with circumferential neck measurements (ICC>0.9). Part of the reason might be that the neck circumference measurements are easy to perform and require less anatomical skill to identify the landmarks. Moreover, because of the shape of the neck, the circumferences do not vary much when the tapes are slightly misplaced from the optimal position. However, the precision of neck circumference measurements was lower than that achieve when measuring shorter distances.

The variation in precision and reliability can be explained by the following factors. First, several facial landmarks are difficult to pinpoint precisely, especially in obese subjects and patients with severe edema. Second, the curvature of the facial structures, hair, and facial hair make placing the tape close to the skin difficult. Third, facial and neck movements in subjects during measurements could shift the location of landmarks such as the lip angle, eye corners, and crown. Finally, a discrepancy between evaluators could arise if the tape is pulled at different degrees of tension.

We could not determine the optimal tape measured parameter in this study as the reliability and precision of each measure varied significantly. Within-subject deviations should be evaluated in order to compare measurements within a single subject, while ICC values are used to determine the variation in measurements between subjects and those caused by measurement errors [3].

## Limitations

There are a few limitations to this study. This study could not assess the ability of the tape measurement to detect changes in the volume of lymphedema due to the cross-sectional nature of this study. Moreover, it is beyond the scope of this study to state whether tape measurement is an acceptable technique for head and neck lymphedema assessment, as it only compares the precision and reliability between measurements. In addition, the evaluators in this study had varying clinical backgrounds as both physicians and nurses were included. Thus, the results may have varied if the evaluators have similar clinical backgrounds and experience.

## Conclusions

This study reported various values of precision and reliability between tape measurements of point-to-point distances of facial landmarks and circumferences of the head and neck. No tape measurement achieved excellent precision and reliability. Thus, clinicians should not rely on a single measurement when evaluating head and neck lymphedema.

## Supporting information

**S1 Dataset. Subject datasets.**
(XLS)

## Author Contributions

**Conceptualization:** Adit Chotipanich.

**Data curation:** Nampheng Kongpit.

**Formal analysis:** Adit Chotipanich.

**Investigation:** Nampheng Kongpit.

**Methodology:** Adit Chotipanich.

**Project administration:** Adit Chotipanich.

**Writing – original draft:** Adit Chotipanich.

**Writing – review & editing:** Adit Chotipanich.

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
