## [Decision Letter · Decision Letter 0]

11 Mar 2020

PONE-D-20-02964

Precision and reliability of tape measurements in the assessment of head and neck lymphedema

PLOS ONE

Dear Mr Chotipanich,

Thank you for submitting your manuscript to PLOS ONE. After careful consideration, we feel that it has merit but does not fully meet PLOS ONE’s publication criteria as it currently stands. Therefore, we invite you to submit a revised version of the manuscript that addresses the points raised during the review process.

We would appreciate receiving your revised manuscript by Apr 25 2020 11:59PM. To enhance the reproducibility of your results, we recommend that if applicable you deposit your laboratory protocols in protocols.io, where a protocol can be assigned its own identifier (DOI) such that it can be cited independently in the future. For instructions see: http://journals.plos.org/plosone/s/submission-guidelines#loc-laboratory-protocols

We look forward to receiving your revised manuscript.

Kind regards,

Peter Dziegielewski, MD, FRCSC

Academic Editor

PLOS ONE

Journal Requirements:

2. Thank you for including your ethics statement:  "The study’s protocol was approved by the appropriate hospital ethics committee, and all participants provided written informed consent prior to their enrollment. (IRB no. 2/2019)".   

3. Please include a caption for figure 1.

Reviewers' comments:

Reviewer's Responses to Questions

**Comments to the Author**

1. Is the manuscript technically sound, and do the data support the conclusions?

Reviewer #1: No

Reviewer #2: Partly

2. Has the statistical analysis been performed appropriately and rigorously? 

Reviewer #1: Yes

Reviewer #2: Yes

3. Have the authors made all data underlying the findings in their manuscript fully available?

Reviewer #1: No

Reviewer #2: Yes

4. Is the manuscript presented in an intelligible fashion and written in standard English?

Reviewer #1: Yes

Reviewer #2: Yes

5. Review Comments to the Author

Reviewer #1: This study aims to evaluate the precision (standard deviation of measurements of the same subject) and reliability (intraclass correlation coefficient) of tape measurement techniques performed by different measurers in the assessment of head and neck lymphedema. It addresses an important complication of treatment among head and neck cancer patients, impacting on post-treatment morbidity as well as the patient’s quality of life. I would appreciate though if more details were shared by the authors namely:

1. The clinical profile of the diseased subjects (e.g., histopathologic type, stage and neck node involvement, kinds of surgery undergone by each, particularly if neck dissection was done, radiotherapy, etc.). Ideally one would like to see a spectrum of lymphedema that approximates real life practice in order to gauge the directness and applicability of the study. I am also clear how the diagnosis of lymphedema was made in the diseased subjects.

2. The training which the measurers underwent and probably the pre and post training reliability as a measure of the effectiveness of the training

3. The actual tape measures used (are these medical grade? Were they the same for all measurers and how were the tape measures cared for, stored), and what time of the day were the measurements made (lymphedema may worsen later in the day after prolonged upright position)

4. How were the facial and neck landmarks site marked? How precisely were these identified? And how were they kept similar for each measurer? The differences may be due to site marking variations.

Why the authors got the results that they obtained were not adequately explained. For example, the mandibular angle appears to be associated with high standard deviations and low ICCs. Why is this so?

Finally it is very possible that no single measurement can identify significant lymphedema but would a combination of measurements do the job better? The statistical analysis does not appear to show this.

Reviewer #2: This study evaluates various precision and reliability tape measurements of the facial landmarks and circumferences of the head and neck in patients with lymphedema as well as controls. While this study does have a sizeable n, there are few shortcomings to the study. Additionally , studies similar to this one have already been published before, although not cited within the manuscript. (Purcell A, Nixon J, Fleming J, McCann A, Porceddu S. Measuring head and neck lymphedema: The "ALOHA" trial. Head Neck. 2016 Jan;38(1):79-84.,

Nixon J, Purcell A, Fleming J, McCann A, Porceddu S. Pilot study of an

assessment tool for measuring head and neck lymphoedema. Br J Community Nurs.

2014 Apr;Suppl:S6, S8-S11.

While it is recommended that this study, after thorough revision mainly for grammar, is resubmitted for consideration, this manuscript would be better suited for publication in a more subspecialty specific journal like Head and Neck, Otolarygnology White journal, etc.

General comments:

It is a good practice to use active voice instead of passive voice which you have throughout the manuscript. As an example:

The study’s protocol was approved by the appropriate hospital ethics committee.

Instead you should write -> The appropriate hospital ethics committee approved the study’s protocol.

There are multiple commas missing throughout the article, especially before ‘and’ and ‘but’, please correct

There are multiple articles missing throughout the manuscript, please see few examples below and take time to correct the manuscript:

affect reliability of -> affect the reliability of

method in clinical assessment -> method in the clinical assessment

no study related to precision and reliability -> no study related to the precision and reliability

The objective of this study is to evaluate the precision -> The objective of this study is to evaluate the accuracy.

-It is a good practice to not repeat the same words within the same sentence or right after the sentence that used the word before, instead choose synonyms like example provided.

Methods:

From these 20 measurers -> From these 20 measures

6. PLOS authors have the option to publish the peer review history of their article (what does this mean?). If published, this will include your full peer review and any attached files.

Reviewer #1: Yes: Jose M. Acuin

Reviewer #2: No

---

## [Author Response · Author response to Decision Letter 0]

20 Mar 2020

We thank the reviewers for the time and effort that they invested into the review of our manuscript, and for their helpful comments and suggestions.

Reviewer #1: 

This study aims to evaluate the precision (standard deviation of measurements of the same subject) and reliability (intraclass correlation coefficient) of tape measurement techniques performed by different measurers in the assessment of head and neck lymphedema. It addresses an important complication of treatment among head and neck cancer patients, impacting on post-treatment morbidity as well as the patient’s quality of life. I would appreciate though if more details were shared by the authors namely:

1. The clinical profile of the diseased subjects (e.g., histopathologic type, stage and neck node involvement, kinds of surgery undergone by each, particularly if neck dissection was done, radiotherapy, etc). Ideally one would like to see a spectrum of lymphedema that approximates real life practice in order to gauge the directness and applicability of the study. I am also clear how the diagnosis of lymphedema was made in the diseased subjects.

Reply: Agreed. We have added the detail of patients and tumors in the result section (table1, page 7). 

2. The training which the measurers underwent and probably the pre and post training reliability as a measure of the effectiveness of the training

Reply: All measurers must successfully completed practice measurement training with the principle researchers. The measurers consisted of physicians and nurses who must have an experience in working with patients in head and neck clinic for at least 2 years. The measurers were also randomly assigned to minimize biases. However, there could be variation between measurer. We have added more details in the material and method section (page4).

3. The actual tape measures used (are these medical grade? Were they the same for all measurers and how were the tape measures cared for, stored), and what time of the day were the measurements made (lymphedema may worsen later in the day after prolonged upright position)

Reply: We used the same soft vinyl medical measuring tape for every measure. The tape was cleaned with alcohol after use. The time to conduct the measurement depended on convenience of the subjects. However, the measures were conducted consecutively. Thus, the variation from prolonged upright position was minimal. We have added more details regarding the measurement in the material and method section (page 4).

4. How were the facial and neck landmarks site marked? How precisely were these identified? And how were they kept similar for each measurer? The differences may be due to site marking variations.

Reply: In the process of qualification, the measurers must correctly identified landmarks in 1 or 2 model subjects. We agreed with the reviewers that site marking variation caused the difference between measurements. We have discussed the factors that might contribute to the variation. We have added the discussion about a possible error from the measurers in the limitation (page12).

Why the authors got the results that they obtained were not adequately explained. For example, the mandibular angle appears to be associated with high standard deviations and low ICCs. Why is this so?

Reply: Reliability (ICC) generally represents a ratio of true variance (mean square between groups, MSB) over true variance plus error variance (mean square within groups). The facial landmark measurement had narrow standard deviation (low MSW) and MSB was also low. Thus the ICC values of the facial landmark measurement were relatively moderate. On the other hand, the neck circumference measurement had slightly wider standard deviation but the MSB values were much higher. So the ICC values were relatively high. We have added the MSB values in table2 (page8) and the discussion about the interpretation of these results in the discussion section (page11).

Finally it is very possible that no single measurement can identify significant lymphedema but would a combination of measurements do the job better? The statistical analysis does not appear to show this.

Reply: We have performed analysis of combination between 2 measurements with the best precision and reliability. The result was shown in table2. The combination did not improve precision and the reliability was the same.

Reviewer #2: 

This study evaluates various precision and reliability tape measurements of the facial landmarks and circumferences of the head and neck in patients with lymphedema as well as controls. While this study does have a sizeable n, there are few shortcomings to the study. Additionally, studies similar to this one have already been published before, although not cited within the manuscript. (Purcell A, Nixon J, Fleming J, McCann A, Porceddu S. Measuring head and neck lymphedema: The "ALOHA" trial. Head Neck. 2016 Jan;38(1):79-84.,Nixon J, Purcell A, Fleming J, McCann A, Porceddu S. Pilot study of an assessment tool for measuring head and neck lymphoedema. Br J Community Nurs.

2014 Apr;Suppl:S6, S8-S11.)

Reply: Thank you for this suggestion. The results of these studies are in agreement with our results. We have added the references in the discussion section (page11).

While it is recommended that this study, after thorough revision mainly for grammar, is resubmitted for consideration, this manuscript would be better suited for publication in a more subspecialty specific journal like Head and Neck, Otolarygnology White journal, etc.

General comments:

It is a good practice to use active voice instead of passive voice which you have throughout the manuscript. As an example: The study’s protocol was approved by the appropriate hospital ethics committee. Instead you should write -> The appropriate hospital ethics committee approved the study’s protocol. 

There are multiple commas missing throughout the article, especially before ‘and’ and ‘but’, please correct There are multiple articles missing throughout the manuscript please see few examples below and take time to correct the manuscript: affect reliability of -> affect the reliability of method in clinical assessment -> method in the clinical assessment no study related to precision and reliability -> no study related to the precision and reliability. The objective of this study is to evaluate the precision -> The objective of this study is to evaluate the accuracy.

-It is a good practice to not repeat the same words within the same sentence or right after the sentence that used the word before, instead choose synonyms like example provided.

Methods:

From these 20 measurers -> From these 20 measures

 Reply: In response to the reviewer’s comment, we have had this manuscript re-edited by a professional language editor.

---

## [Decision Letter · Decision Letter 1]

5 May 2020

Precision and reliability of tape measurements in the assessment of head and neck lymphedema

PONE-D-20-02964R1

Dear Dr. Chotipanich

We are pleased to inform you that your manuscript has been judged scientifically suitable for publication and will be formally accepted for publication once it complies with all outstanding technical requirements.

With kind regards,

Peter Dziegielewski, MD, FRCSC

Academic Editor

PLOS ONE

Additional Editor Comments (optional):

Reviewers' comments:

Reviewer's Responses to Questions

**Comments to the Author**

1. If the authors have adequately addressed your comments raised in a previous round of review and you feel that this manuscript is now acceptable for publication, you may indicate that here to bypass the “Comments to the Author” section, enter your conflict of interest statement in the “Confidential to Editor” section, and submit your "Accept" recommendation.

Reviewer #2: All comments have been addressed

2. Is the manuscript technically sound, and do the data support the conclusions?

Reviewer #2: Yes

3. Has the statistical analysis been performed appropriately and rigorously? 

Reviewer #2: Yes

4. Have the authors made all data underlying the findings in their manuscript fully available?

Reviewer #2: Yes

5. Is the manuscript presented in an intelligible fashion and written in standard English?

Reviewer #2: Yes

6. Review Comments to the Author

Reviewer #2: After a thorough and extensive revision of the manuscript, the content presented reads efficiently and in a grammatically sound presentation. The addressed changes of the text, as well as figures, clarify the presented data inc a transparent manner. Currently, the study has a complete discussion as well as appropriately detailed limitations. Overall all of the included Figures and Tables are necessary and appropriate.

7. PLOS authors have the option to publish the peer review history of their article (what does this mean?). If published, this will include your full peer review and any attached files.

Reviewer #2: No

---

## [Editor Report · Acceptance letter]

8 May 2020

PONE-D-20-02964R1 

Precision and reliability of tape measurements in the assessment of head and neck lymphedema 

Dear Dr. Chotipanich:

I am pleased to inform you that your manuscript has been deemed suitable for publication in PLOS ONE. Congratulations! Your manuscript is now with our production department. 

With kind regards,

on behalf of

Dr. Peter Dziegielewski 

Academic Editor

PLOS ONE